# Importance of Water Transport in Mammalian Female Reproductive Tract

**DOI:** 10.3390/vetsci10010050

**Published:** 2023-01-11

**Authors:** Lluis Ferré-Dolcet, Maria Montserrat Rivera del Alamo

**Affiliations:** 1San Marco Veterinary Clinic and Laboratory, 35030 Veggiano, Italy; 2Departament de Medicina i Cirurgia Animals, Veterinary College, Universitat Autònoma de Barcelona, 08913 Bellaterra, Spain

**Keywords:** aquaporins, female reproductive tract, reproduction, placenta, water homeostasis

## Abstract

**Simple Summary:**

The female reproductive tract undergoes several structural changes during estrus and menstrual cycles, pregnancy and parturition that involve important regulation of the exchange of solutes to prepare the uterus for conception, implantation and fetal development. The expression of a particular aquaporin isoform has specific locations in the reproductive tract that differs among species probably because of the type of uterine anatomy or type of placentation. These locations involve different actions, metabolism and functions. AQPs have been identified, but the regulation of water transportation in the uterus and placenta is poorly understood. The expression and location of aquaporins in the uterus are regulated by ovarian steroid hormones such as estradiol and progesterone, suggesting that these aquaporins play important roles in water and other solutes transportation in uterine imbibition and amniotic fluid resorption.

**Abstract:**

Aquaporins (AQPs) are involved in water homeostasis in tissues and are ubiquitous in the reproductive tract. AQPs are classified into classical aquaporins (AQP0, 1, 2, 4, 5, 6 and 8), aquaglycerolporins (AQP3, 7, 9, and 10) and superaquaporins (AQP11 and 12). Nine AQPs were described in the mammalian female reproductive tract. Some of their functions are influenced by sexual steroid hormones. The continuous physiological changes that occur throughout the sexual cycle, pregnancy and parturition, modify the expression of AQPs, thus creating at every moment the required water homeostasis. AQPs in the ovary regulate follicular development and ovulation. In the vagina and the cervix, AQPs are involved mainly in lubrication. In the uterus, AQPs are mostly mediated by estradiol and progesterone to prepare the endometrium for possible embryo implantation and fetal development. In the placenta, AQPs are responsible for the fluid support to the fetus to maintain fetal homeostasis that ensures correct fetal development as pregnancy goes on. This review is focused on understanding the role of AQPs in the mammalian female reproductive tract during the sexual cycle of pregnancy and parturition.

## 1. Aquaporins and Reproduction

Water is crucial in all processes for living cells and its movement across the plasma membrane is fundamental for water homeostasis during fetal development [1]. In the early 1990s, a 28 kDa protein was found in the red blood cells and renal tubules and named CHIP 28 (Channel-like Integral Protein of 28 kDa) [2]. It was organized in tetramers and was selectively permeable to water thereby, explaining the high permeability of those membranes containing that specific protein [2,3,4]. After cloning and sequencing, the protein was named aquaporin 1 (AQP1) [5,6,7]. Since this discovery, a lot of research has been performed and other proteins with similar characteristics have been discovered and named aquaporins with a sequential number according to the order of discovery. As these proteins have been found in several tissues and cells, they have been considered to play an important role in regulating the water permeability of cell membranes, thus contributing to water homeostasis and osmoregulation. To date, a total of 13 aquaporins (AQP0–12) have been described in mammals and 11 of them (AQP1–9, 11 and 12) have been described to play a role in water movement between capillaries, interstitial and luminal compartments in the female reproductive tract [8].

## 2. Molecular Structure, Function, and Location of Aquaporins (AQPs)

AQPs structure consists of six domains that are connected by five loops with each polypeptide (formed by a single chain with approximately 270 amino acids) presenting terminal amino and carboxyl groups, which are always located in the cell cytoplasm [9]. These loops are either extracellular (loops A, C, and E) or intracellular (loops B and D) [10]. All AQPs are conformed by two very similar halves which are joined by loop C, which has a structural function altogether with loops A and D [11], while B and E loops are essential for water permeability and pore formation (Figure 1 and Figure 2) [12]. AQPs are tetrameric structures [12], although AQP4 can form smaller oligomers.

Aquaporins (AQPs) are small (about 28–35 kDa) hydrophobic integral cellular membrane proteins playing a role in water transport and increasing the permeability of bilayer lipid membranes [13]. Its water permeability is in the order of 3 × 10^9^ water molecules per second for AQP1 and closer values for the other aquaporins [14].

Aquaporins are highly selective to water transport, avoiding the passing of ions and protons [15]. The presence of an Arg-195 residue in the pore prevents the pass of protonated water [16], and a second barrier composed by the intracellular loops with the asparagine-proline-alanine (NPA) sequence reorients the water molecules to facilitate the membrane crossing, disrupting interactions between molecules and preventing them from passing through [17]. In general, these water channels also prevent the passing of other anions because of their 2.8 Å pore measure (less than any hydrated ion). However, in the aquaglyceroporins, an alternative glycine in HIS-180 is associated with a bigger pore with the capacity to transport glycerol and other solutes [18].

Based on their permeability, aquaporins have been classified into three different categories: (1) classical AQPs, (2) aquaglycerolporins and (3) superaquaporins.

Classical aquaporins include AQP0, 1, 2, 4, 5, 6 and 8. They are considered only water-selective channels. Nevertheless, AQP6 is considered to be able to transport ions [10] and AQP8 is also permeable to ammonia [11].

Aquaglycerolporins include AQP3, 7, 9, and 10. In addition to being permeable to water, these AQPs are also permeable to urea and glycerol. Furthermore, AQP9 has the facility to transport also monocarboxylates, purines and pyrimidines [12].

Superaquaporins include AQP11 and 12. These AQPs are located in the cytoplasm and its permeability has not been fully determined yet [14].

AQPs are regulated by intracellular factors like pH and phosphorylation, mainly mediated by protein kinase A for its phosphorylation and regulation of the water balance mechanism [9]. On the other hand, the function of most AQPs can be inhibited by mercury (Hg^2+^) because of a cysteine residue (Cys-189) in the E loop. However, AQP4 and 6 are activated by Hg^2+^ [10].

AQPs are ubiquitous in mammals and usually are not restricted to a unique tissue. However, their function will be determined by their specific location in the cell or organ [15]. It has been shown that several organs such as the brain, lungs, muscles, eye, ear, skin, adipose tissue, testis, uterus and placenta contain more than one AQP [16,17,18,19,20,21,22,23].

First studies in AQPs were performed in kidneys [24]. The kidney contains approximately one million nephrons that filter around 180 L of plasma every day, in addition to water and other solutes reabsorption [25]. Thereafter, AQPs have been found in many other tissues and organs including the lungs, pancreas, brain, gastrointestinal tract, eye, ear, immune system, skin, adipose tissue, muscles and in different parts of the genital tract (Table 1) [16,17,26,27,28,29]. Many clinical studies have shown that the failure of the function of AQPs results in pathologies [30,31,32], leaning on the importance of water trafficking for all biological processes. Specifically focusing on reproduction, it has been demonstrated that some aquaporins are regulated by sex steroid hormones [33,34,35].

## 3. AQPs in the Female Reproductive Tract

### 3.1. AQPs in the Ovary

AQP1 is expressed in the capillaries and vessels of the sow ovary during the end of the follicular phase, while AQP5 has been described to be present in the primordial follicles and granulosa cells, the last ones also expressing AQP9 [67]. A large volume of water is required for follicle expansion during follicular growth and development. In rats, this water movement into the follicular antral cavity seems to be mediated by the granulosa cells that express AQP7, 8 and 9 [63]. Moreover, a significant increase in the expression of both AQP2 and 3 has been observed in the theca and granulosa cells of human pre-ovulatory follicles during the phase of ovulation, suggesting that this mechanism may be implicated in the follicular rupture, whereas AQP1 showed an increased expression in the late ovulatory and postovulatory period and AQP4 expression decreased prior to ovulation [39].

### 3.2. AQPs in the Vagina and Cervix

Several aquaporins were described to be present in both the mammalian vagina and cervix where they contribute to lubrification for mating, sperm movement and entrance into the uterus, closure of the cervix during pregnancy and opening for delivery [36].

In the rat vaginal epithelium, AQPs0–3 and AQPs10–12 expression has been described, and AQP0 is apparently exclusive of rodent vagina [36]. Regarding AQP1, as it has been shown in different organs, seems to be mostly expressed on the vascular endothelia but, together with AQP2, shows a translocation from the intracellular membranes of the epithelium to the plasma membrane when the pelvic nerve is stimulated, thus contributing to the vaginal lubrification and the increase of vaginal blood flow [36,47,68]. AQP1 has also been described in vaginal smooth muscle, suggesting that the vagina is a “water rapid-flow” organ either out or into the muscular cells [48]. On the other hand, AQP3 expression has been shown to be specific to the plasma membrane of the epithelial vaginal cells with its function described as the lubrification of the vaginal canal.

Rodent vaginal epithelium also expresses AQPs10–12. A previous study showed that these AQPs are under-expressed in ovariectomized rats, suggesting that their expression might be mediated by estrogens [36]. In addition to AQP3, the cervix of rodents expressed AQPs4, 5 and 8, all of which change location during pregnancy from the apical cell layers of the cervical epithelia to the cell membrane and cytoplasm of the basal cervical epithelial cells [36].

AQP3 is less expressed in basal cells of the cervical epithelia in mice. However, its expression peaks in pre-parturient subjects [58,59]. Unlike, AQPs4, 5 and 8 are expressed in the apical cervical cell layers with their expression also significantly increased during the last 12 days of pregnancy, probably due to the endocrinal effect of relaxin [58]. The canine cervix showed immunoreactivity to AQP5 in the epithelial luminal surface during diestrus, with cytoplasmatic and cell membrane staining. When the cervix was evaluated during low progesterone concentrations stages, no AQP5 expression was detected [33].

In women, in addition to AQPs1–3, AQP5 and 6 are expressed in the cytoplasm of the vaginal epithelium, suggesting a possible role in lubrification [40,56]. Unfortunately, human cervical expression of aquaporins (AQP3 and 8) has been only studied under pathologic conditions such as cervical cancer or cervicitis [41].

### 3.3. AQPs in the Uterus

The expression of AQPs, specifically AQP1, 2, 5, 8 and 9, are regulated by progesterone and estradiol [49,51,54,69]. The uterine distribution of AQP1, 2, 3 and 8 isoforms suggest a role in water flow during uterine imbibition [33,35,54,55,62]. There is evidence of the role of AQPs in the sexual cycle where they induce edema and glandular secretion in the uterus [49,51,54,69]. This mechanism is considered a special diffusion of fluids that occurs in the mammalian uterus during the estrous cycle and pregnancy when the uterus increases its volume due to the absorption of water and solids-colloids. This mechanism is mediated by steroid hormones producing local changes such as edema, transcellular and intraluminal fluid movement, hyperemia and higher capillary permeability [70,71,72,73]. Clemenston et al. (1997) described that estradiol-17α induces the secretion of water and other substances, such as sodium and potassium, into the lumen of uterine horns in ovariectomized rats, while progesterone was responsible for the reabsorption of these substances.

During the follicular phase of the sow (days 17 and 19 of estrus cycle), AQP1, 5 and 9 are detected in the endometrial vessels, myometrial and smooth muscle cells, and glandular and luminal uterine epithelial cells respectively [67].

The equine endometrium expresses several aquaporins (AQP0–5 and AQP7–12), showing some variations in its expression throughout the estrus cycle and early pregnancy [35]. In the latest study, endometrial samples from mares were obtained during anestrus, estrus, day 8 of the cycle, day 14 of the cycle and day 14 of pregnancy. AQP0, 2 and 5 location was evaluated by immunochemistry, showing expression in luminal and glandular epithelial cells together with immunoreactivity in the stromal cells with significantly higher expression of AQP0 in samples evaluated after the 14th day of the cycle in both cyclic and pregnant mares. For AQP2, immunoreactivity was stronger in the luminal epithelial cells after day 14 of the cycle, whereas AQP5 showed stronger immunoreactivity on day 8 of the cycle in the glandular epithelium. When RT-PCR for AQP expression in the mare endometrium was performed, AQP0 showed its higher expression on day 14 of the cycle, as it happens with AQP2 and AQP12. On the other hand, AQP1, 4, 8, 9 and 11 presented their highest expression on day 14 of pregnancy. AQP10 was expressed during all the stages of the sexual cycle but showed a lower expression during estrus. Lastly, AQP3 and AQP7 showed their maximum expression during the 8th day of the estrus cycle [35].

To describe the mechanism that balances water flow in the uterus, Jablonski et al. (2003) studied the expression and function of some AQPs in the ovariectomized mouse uterus treated with sexual serum steroid hormones. The study describes that AQP1, 2, 3 and 8 might participate in water movement during uterine imbibition. Myometrial AQP1 is suggested to be slightly regulated by ovarian steroid hormones, but its actual mechanism has not been elucidated [51]. AQP2 was found to be strongly upregulated by estrogen in epithelial cells and myometrium of the uterus, and AQP3, similarly, was also detected in uterine epithelial cells, suggesting a contribution to water flow into the uterine lumen. In addition, He et al. (2006) described that human endometrial AQP2 was menstrual cycle-dependent because, during the mid-secretory phase, its expression was increased and showed a positive correlation with progesterone and estrogens [69]. On the other side, AQP8 was also found in the myometrium but was the only one found in the stroma of the uterus suggesting that water moves from the myometrium to the lumen, protecting the myometrial layer from edema [51].

On the other hand, Lindsay and Murphy (2006) reported that progesterone up-regulates AQP5 in the rat uterus [50].

In Carnivores, Aralla et al. (2009) showed that the dog endometrium expressed AQP1, 2 and 5. In cycling bitches, AQP1 was expressed in the vascular endothelia and in the circular layers of the myometrium. AQP2 was expressed in the apical membrane of the glandular epithelium cells during proestrus and estrus, whereas no expression was detected during diestrus when high progesterone levels were present. Nevertheless, in the smooth myometrial cells, AQP2 expression was also present throughout the whole estrus cycle, suggesting that it is not dependent on progesterone secretion. On the other hand, AQP5 is expressed in the endometrial apical cells of the glandular and luminal epithelia only during the diestrus phase, when high progesterone levels were present [33]. On the other side, pregnant and pseudopregnant queens showed a change in AQP2 and 8 locations from the cytoplasm to the cytoplasmatic cell membrane, suggesting a contribution of these aquaporins in both embryo implantation and development [54].

Once ovulation has occurred, the oocyte needs to be transported to the uterus across the isthmus by muscular contractions, edema and vascular distension [50,60]. In the rat epithelial cells of the oviduct, AQP5, 8 and 9 have been described to be present, suggesting a role in the production of oviductal fluid which, along with steroid hormones, may regulate fertilization and early embryo development [62].

Lindsay and Murphy (2004) described that the increase of AQP1 expression in the mesometrial muscle contributes to the antimesometrial positioning of the embryo in the uterine lumen. The same authors described that AQP5 reorganizes its expression from a completely cytoplasmic location during the first days of pregnancy, to a predominant apical plasma membrane organization at the time of implantation in the rat uterus. This fact would explain the pathway for uterine luminal fluid reduction at the time of implantation, just in collaboration with AQP1 and AQP4.

Likewise, He et al. (2006) described a high expression of AQP2 at the mid-secretory phase of human endometrium, suggesting a role for embryo receptivity. All these facts suggest that AQPs might regulate tissue fluid balance during implantation.

### 3.4. AQPs in the Placenta

It is well known that of all the solutes that are needed during pregnancy for fetal growth, water is without any doubt the most important [74]. As water requirements increase during pregnancy due to an increase in fetal weight [57], it seems logical to think that AQPs might be involved in the maintenance of pregnancy and fetal development. Many authors [52,64,75,76] have described the involvement of AQPs in amniotic fluid volume control because of its location in the different fetal membranes. This fact results in a maternal-fetal water flux critical for normal fetal growth [52]. The expression of a specific AQP isoform has a particular location in both uterus and placental transfer zone that differs among species because of the type of uterine anatomy or type of placentation. This fact may give different sites of action, metabolic action and roles to these proteins.

Once the fetus starts to grow and develop, amniotic fluid is needed for the creation of a fluid-filled compartment that will be essential. To date, seven AQPs (AQP1, 2, 3, 5, 8, 9 and 11) among the 13 identified in mammals have been described in mammalian chorionic membranes and placenta.

The presence of AQP1 in the vascular endothelium of the chorion of different species has been described by many authors, suggesting water flow to the embryo during pregnancy [38,43,44]. In fact, AQP1 is expressed in ovine syncytiotrophoblasts and chorionic endothelium [23] and in human chorion and amnion cytotrophoblasts [42]. AQP1 was also detected in placental blood vessels in queens and human beings [43,54].

AQP2 is expressed in the chorionic syncitiotrophoblasts and cytotrophoblasts of queen and human placenta suggesting an important role in fetal development [54,56].

AQP3 has been described in the ovine and human chorion and placenta but without expression in the amniotic epithelium [23,44]. AQP3 is also expressed in the dog and cat chorionic cells of the placental labyrinth [38,54] and in the amnion and yolk sacs of canines [38]. AQP3 is apparently the most highly expressed AQP in the placenta with a similar expression to AQP8 in the ovine and feline trophoblastic and epithelial cells [37,54].

AQP4 is present in both human syncytiotrophoblast and endothelial cells and stroma of placental villi [61] from the first to the third trimester of pregnancy. Also, Escobar et al. (2012) described AQP4 and 5 expression in human chorionic villi samples from the 10th to 14th week in normal pregnancies [45]. In addition, in canine species, during pregnancy, AQP5 seems to be expressed in amniocytes and the columnar cells of the allantochorion [74].

As the urea produced by the fetus must be removed from the fetal circulation to the maternal circulation, it is interesting to think that AQP8 may be involved in the urea transport across the fetal membranes. Concretely, AQP8 was first described to be present in the epithelial cells of chorion and amnion and in the syncytiotrophoblasts of the human placenta [66] and in the rat placenta, making it permeable to water and urea but not to glycerol [65]. Similar results were found in canine, feline and mice placenta [38,53,54]. In addition to AQP3, both proteins have been described to be involved in water imbibition for the preparation of the endometrium for a possible pregnancy [46,51,54,77].

Damiano et al. (2001) described by RT-PCR, immunoblotting and immunochemistry the presence of AQP9 in the apical membranes of syncytiotrophoblasts of human term placenta. Further studies realized by Wang et al. (2005) reported AQP9 mRNA expression in ovine amnion and allantois, indicating to be a major water channel for intramembranous amniotic fluid reabsorption.

The first time that AQP11 was detected in fetal membranes was in a study by Escobar et al. (2012) who revealed its expression in the chorionic villi between the 10th and 14th week of human pregnancy. Moreover, Prat et al. (2012) also described AQP1 mRNA and protein in human chorion and amnion.

## 4. Conclusions

Many studies have described the importance of water movement across both male and female reproductive tracts for gamete formation and fertilization, ovulation, oocyte transport and amniotic fluid balance during pregnancy. Aquaporins are structural proteins involved in cellular water trafficking transportation of different tissues and organs. AQPs have been identified in the female reproductive tract of various mammalian species. It is true that the regulation of water across the uterus and the placenta is a clue for reproductive functions prior to ovulation until parturition and uterine involution. Thus, the expression and function of water transporters in the male and female reproductive tract warrant further investigation. The expression and location of AQPs in the uterus are regulated by estradiol and progesterone, suggesting that these AQPs play important roles in reproduction.

## Figures and Tables

**Figure 1 vetsci-10-00050-f001:**
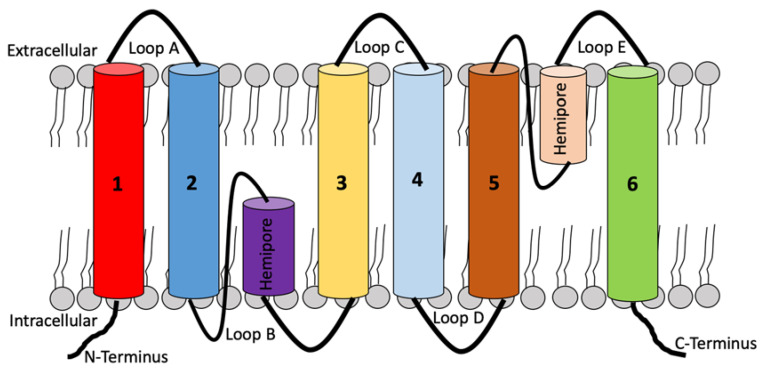
Two-dimensional structure of AQP1 based on six transmembrane pores and two hemi pores helices.

**Figure 2 vetsci-10-00050-f002:**
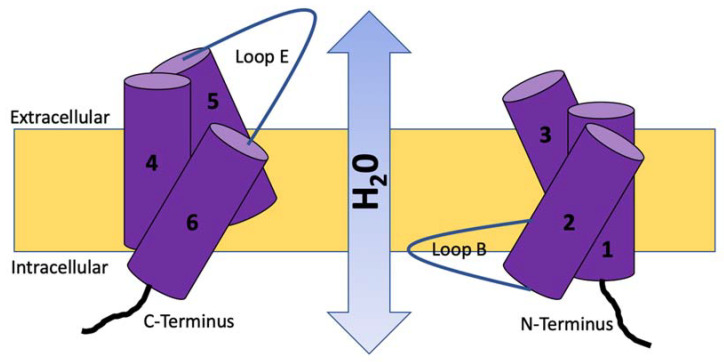
Two-dimensional structure of AQP1 showing the water passage across the pore.

**Table 1 vetsci-10-00050-t001:** Localization of AQPs in the female reproductive tract. Detection method is specified for each aquaporin in each species.

Aquaporin	Species	Localization	Analytic Method	References
AQP0	Horse	Endometrial luminal, glandular epithelial and stromal cells	RT-PCR, WB, IHC	[35]
	Rat	Cytoplasm of vaginal epithelia	WB, IHC	[36]
AQP1	Sheep	Endothelium of capillaries under the trophoblast layer of the chorion and maternal capillary endothelium of cotyledons	RT-PCR, WB, IHC	[23,37]
	Dog	Vascular endothelia, glandular epithelium of the endometrium, smooth muscle cells of myometrium	WB, IHC	[33,38]
	Horse	Endometrium	RT-PCR	[35]
	Human	Amniotic and chorionic epithelia and villi. Placenta. Uterus. Large, rounded cells of the theca, granulosa cell layer. Capillaries and venules of vagina and cervix.	RT-PCR, WB, IHC	[39,40,41,42,43,44,45,46]
	Rat	Cytoplasm and plasma membrane of myometrium. Capillaries and venules of vaginal lamina propia	WB, IHC	[47,48,49,50]
	Mouse	Myometrium. Vessel walls of placental labyrinth and yolk sac	RT-PCR, WB, IHC	[51,52,53]
	Cat	Endometrial endothelia, glandular epithelium and myometrium	WB, IHC	[54]
	Pig	Ovarian, oviductal and uterine capillary epithelium	IHC	[55]
AQP2	Dog	Apical membrane of glandular epithelium of the uterus, smooth muscle cells of myometrium and blood vessel musculature	WB, IHC	[33]
	Horse	Endometrial luminal, glandular epithelial and stromal cells. Cytoplasm of vaginal epithelium	RT-PCR, WB, IHC	[35,40]
	Human	Uterus. Large rounded cells of the theca, granulosa cell layer. Fetal membrane, trophoblastic cells and syncytiotrophoblsts	RT-PCR, IHC	[39,45,46,56]
	Rat	Superficial layer of vaginal epithelium	WB, IHC	[47]
	Mouse	Epithelial endometrial cells and myometrium. Fetal membranes and placenta	RT-PCR, WB, IHC	[51,52]
	Cat	Endometrial cells of luminal and glandular epithelia. Syncytiotrophoblasts and cytotrophoblasts	WB, IHC	[54]
AQP3	Sheep	Apical membrane of trophoblastic cell layer of villous chorion and fetal trophoblastic cell layer within chorionic villi	RT-PCR, WB, IHC	[23,37]
	Horse	Endometrium	RT-PCR	[35]
	Human	Amiotic and chorionic epithelia and villi. Placenta. Uterus. Granulosa and theca cell layers. Plasma membrane of vaginal epithelium. Squamous cervical epithelia	RT-PCR, WB, IHC	[39,40,41,42,43,44,45,57]
	Rat	Plasma membrane and cytoplasm of vaginal epithelia	WB, IHC	[36]
	Mouse	Basal cell layers of cervical epithelium. Epithelial cells of endometrium and myometrium. Trophoblastic cells	RT-PCR, Northern hybridation, in situ hybridation, WB, IHC	[51,52,53,58,59]
	Cat	Endometrial cells of luminal and glandular epithelia. Syncytiotrophoblasts and cytotrophoblasts	WB, IHC	[54]
	Dog	Apical endometrial cell membrane of glandular epithelium. Placental allantochorion. Yolk sac	IHC	[60]
AQP4	Horse	Endometrium	RT-PCR	[35]
	Human	Placenta, chorionic villi and uterus. Granulosa and theca cell layers	RT-PCR, IHC	[39,45,61]
	Mouse	Apical cell layers and mucus-secreting cervical cell surface. Fetal membranes and placental	RT-PCR, Northern hybridation, in situ hybridation, IHC	[52,58]
AQP5	Dog	Apical membrane of glandular and luminal epithelium of the uterus. Cervical luminal epithelia. Allantochorion cells	WB, IHC	[33,38]
	Horse	Endometrial luminal, glandular epithelial and stromal cells	RT-PCR, WB, IHC	[35]
	Rat	Plasma membrane and cytoplasm of vaginal epithelia. Endometrial luminal epithelial cells. Cytoplasm of oviductal epithelial cells	RT-PCR, WB, IHC	[36,49,50,54,62]
	Mouse	Basal and apical cervical epithelium cells. Fetal membranes and placenta	RT-PCR, Northern hybridation, in situ hybridation, IHC	[52,58,59]
	Pig	Flattened follicle cells of primordial follicles and granulosa cells of developing follicles. Muscle layers and luminal epithelial cells of the oviduct and uterus	IHC	[55]
	Human	Placenta, chorionic villi and uterus	RT-PCR	[45]
AQP6	Rat	Plasma membrane and cytoplasm of vaginal epithelia	WB, IHC	[36]
	Human	Cytoplasm of vaginal epithelium	WB, IHC	[40]
	Mouse	Fetal membranes and placenta	RT-PCR, IHC	[52]
AQP7	Horse	Endometrium	RT-PCR	[35]
	Rat	Granulosa cells	ELISA, WB, IHC	[40,63]
	Mouse	Fetal membranes and placenta	RT-PCR, IHC	[52]
	Human	Placenta and uterus. Cytoplasm of vaginal epithelium	RT-PCR, WB, IHC	[45]
AQP8	Horse	Endometrium	RT-PCR	[35]
	Rat	Cytoplasm of oviductal epithelial cells. Granulosa cells	RT-PCR, ELISA, WB, IHC	[62,63]
	Mouse	Basal and apical cervical epithelium cells. Stromal endometrial cells and myometrium. Basal component of fetal membranes. Placenta	RT-PCR, Northern hybridation, in situ hybridation, WB, IHC	[51,52,53,58,64,65]
	Human	Amiotic and chorionic epithelia and villi. Uterus. Squamous cervical epithelia	RT-PCR, WB, IHC	[41,42,45,66]
	Cat	Endometrial cells of luminal and glandular epithelia. Syncytiotrophoblasts and cytotrophoblasts	WB, IHC	[54]
	Pig	Granulosa cells of developing follicles, oviductal myometrium and glandular and luminal epithelium of the uterus	IHC	[55]
	Dog	Placental spongy zone and lining cells of amnion and allantoic sac	IHC	[38]
	Sheep	Trophoblasts	RT-PCR	[37]
AQP9	Horse	Endometrium	RT-PCR	[35]
	Rat	Apical membrane of oviductal epithelial cells. Granulosa cells	RT-PCR, ELISA, WB, IHC	[62,63]
	Human	Amiotic and chorionic epithelia and villi. Uterus	RT-PCR, WB, IHC	[42,45,57]
	Mouse	Fetal membranes and placenta	RT-PCR, IHC	[52,53]
	Dog	Syncytiotrophoblasts	IHC	[38]
AQP10	Horse	Endometrium	RT-PCR	[35]
	Rat	Vagina	WB	[36]
AQP11	Horse	Endometrium	RT-PCR	[35]
	Rat	Capillaries and venules	WB, IHC	[36]
	Human	Amiotic and chorionic epithelia and villi. Uterus	RT-PCR, WB, IHC	[42,45,52]
AQP12	Horse	Endometrium	RT-PCR	[35]
	Rat	Vagina	WB	[36]
	Mouse	Fetal membranes and placenta	RT-PCR, IHC	[43]

## Data Availability

Not applicable.

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
