# Peer review of "Importance of Water Transport in Mammalian Female Reproductive Tract"

_vetsci, 2023, doi:10.3390/vetsci10010050_

Round 1

Reviewer 1 Report

This work is interesting.

Please use suggested comments in the attached pdf to improve.

The Conclusions needs to be rewritten to make it concise.

Author Response

Modifications to the text have been made. English have been controlled by a native speaker. Conclusion has been rewritten.

Reviewer 2 Report

The authors provide a systematic and comprehensive discussion of the importance of aquaporins in the female mammalian reproductive tract, corroborated by multiple citations, but the conclusions still need improvement and it would be interesting to further illustrate the extent to which this importance will influence future relevant studies.

Author Response

Thank you for your help on improving the manuscript. The query has been adjusted with some modifications.

Reviewer 3 Report

Dear editors,

this review summarizes the knowledge about the Aquaporin expression in the female reproductive tract. Specifically, the authors focused on the understanding of the Aquaporin roles in the mammalian female reproductive tract during the sexual cycle, pregnancy and parturition. The topic is very interesting and current. The references are specific and relevant. Ref. 25: in the list of the references, please add the missing information according to the correct format.

Summary and Abstract: They are well written and recap the information contained in the main text without repetitions. However, since the reader may not know about the discussed topic, the journal rules establish that the summary cannot contain abbreviations.

Key-words: they are pertinent and consistent with the topic. You could add the key-word “water homeostasis”  since it is the main function of the Aquaporines.

The manuscript is well structured. However, I have some comments and suggestions:

1.      It would be better to add more detailed figures of the Aquaporins where the channel of their tetrameric structure is well recognizable. The bidimensional figure (Fig. 1) is useful but not enough to clearly represent the water transfer across the membrane. Since it is a review, it would be interesting to add in this figure the potential pathways that can modify the Aquaporin permeability (the main supposed and hypothesized pathways also). I suggest to move all these figures next to the paragraph 2 in order to be more easily searchable.

2.      The Table 1 could be improved. Changing the disposition of the column and row names, I suggest to add the analytic method used by the authors (i.e. PCR, ELISA, IHC...) for each tissue (i.e. cervix, uterus…) in which the specific Aquaporin has been found. In addition, it would be important to specify in which cells these molecules are expressed (es. muscular, endotelial, granulosa cells …). Some of these information are available in the main text but in a review, since it describes the state-of-the-art of a research topic,, it would be suitable to find a very detailed summary table.

3.      In my opinion, it is necessary to ague more about the findings of other authors, extrapolating more information and interpretations from the mentioned papers. The number of the references is remarkable, so I would have expected a longer text. Specifically, the paragraph “3.4. AQPs in the placenta” could be improved, since it seems to consist just of a list of findings. It is therefore advisable to add some comments, some of them are presented in the conclusions. The conclusion paragraph doesn not need to be that long or contain very detailed information or references (these information should be discussed in the main text). In the conclusion section, the state-of-the-art of the topic is summarily presented, without repetitions with the abstract or summary. Moreover, the potential future applications of these new knowledge could be expressed in this section.

4.      Finally, I suggest to improve the written expression diversifying the syntactic constructs: i.e. an excessive use of the “suggesting a” form could make the text repetitive.

Overall the paper is very captivating and, since it adds useful information in the reproduction field, it deserves to be published after minor revisions.

Author Response

Thank you very much for your effort in improving the manuscript. Our response is attached.

Round 2

Reviewer 1 Report

Accept